# CRISPR/Cas9-Based Knock-Out of the *PMR4* Gene Reduces Susceptibility to Late Blight in Two Tomato Cultivars

**DOI:** 10.3390/ijms232314542

**Published:** 2022-11-22

**Authors:** Ruiling Li, Alex Maioli, Zhe Yan, Yuling Bai, Danila Valentino, Anna Maria Milani, Valerio Pompili, Cinzia Comino, Sergio Lanteri, Andrea Moglia, Alberto Acquadro

**Affiliations:** 1Plant Genetics and Breeding, Department of Agricultural, Forest and Food Science (DISAFA), University of Torino, 10095 Grugliasco, Italy; 2Plant Breeding, Wageningen University & Research, 6708 PB Wageningen, The Netherlands

**Keywords:** CRISPR/Cas9, PMR4 gene, susceptibility, *Solanum lycopersicum* L., *Phytophthora infestans*

## Abstract

*Phytophthora infestans*, the causal agent of late blight (LB) in tomato (*Solanum lycopersicum* L.), is a devastating disease and a serious concern for plant productivity. The presence of susceptibility (S) genes in plants facilitates pathogen proliferation; thus, disabling these genes may help provide a broad-spectrum and durable type of tolerance/resistance. Previous studies on *Arabidopsis* and tomato have highlighted that knock-out mutants of the *PMR4* susceptibility gene are tolerant to powdery mildew. Moreover, *PMR4* knock-down in potato has been shown to confer tolerance to LB. To verify the same effect in tomato in the present study, a CRISPR–Cas9 vector containing four single guide RNAs (sgRNAs: sgRNA1, sgRNA6, sgRNA7, and sgRNA8), targeting as many *SlPMR4* regions, was introduced via *Agrobacterium*-*tumefaciens*-mediated transformation into two widely grown Italian tomato cultivars: ‘San Marzano’ (SM) and ‘Oxheart’ (OX). Thirty-five plants (twenty-six SM and nine OX) were selected and screened to identify the CRISPR/Cas9-induced mutations. The different sgRNAs caused mutation frequencies ranging from 22.1 to 100% and alternatively precise insertions (sgRNA6) or deletions (sgRNA7, sgRNA1, and sgRNA8). Notably, sgRNA7 induced in seven SM genotypes a −7 bp deletion in the homozygous status, whereas sgRNA8 led to the production of fifteen SM genotypes with a biallelic mutation (−7 bp and −2 bp). Selected edited lines were inoculated with *P. infestans*, and four of them, fully knocked out at the *PMR4* locus, showed reduced disease symptoms (reduction in susceptibility from 55 to 80%) compared to control plants. The four SM lines were sequenced using Illumina whole-genome sequencing for deeper characterization without exhibiting any evidence of mutations in the candidate off-target regions. Our results showed, for the first time, a reduced susceptibility to *Phytophtora infestans* in pmr4 tomato mutants confirming the role of KO *PMR4* in providing broad-spectrum protection against pathogens.

## 1. Introduction

Tomato (*Solanum lycopersicum* L.) is the most cultivated and consumed vegetable worldwide, with more than 186.8 million tons harvested in 2020. Its gross production value rose from approximately 32.9 billion US$ in 2000 to 92.8 billion US$ in 2019 (FAOSTAT, https://www.fao.org/faostat (accessed on 4 February 2022). The global demand for tomato has considerably increased in recent years because of its various uses in raw, cooked, and processed food, as well as its substantial contribution to the human diet, as it is rich in lycopene, vitamins, and minerals. Tomato also represents a model plant for biological research because of its short life cycle, chromosomal ploidy (2n = 2x), genome sequence availability, and amenability to transformation methodologies [1,2].

Until the 1950s, tomato breeders developed multipurpose cultivars to meet agricultural and market demands. Subsequently, breeding focused on developing cultivars that were specifically suitable for industrial transformation [3] or fresh markets [4]. The differentiation between fresh and processed accessions reflects all breeding efforts to provide welcomed, commercial tomato cultivars [5]. However, highly selected tomato cultivars are affected by many pathogens, such as viruses, viroids, fungi, oomycetes, bacteria, and nematodes, which cause drops in production and affect fruit quality [6]. Among them, late blight (LB), caused by the etiological agent *Phytophthora infestans*, is a devastating disease and a serious concern for plant productivity [7], as it can destroy an entire unprotected tomato crop within 7–10 days of infection [8]. The main route for disease control is fungicide application, which is an onerous practice with a high environmental impact and contributes to the selection of resistant isolates [9]. The development of cultivars with reduced susceptibility to LB, as well as other pathogens, represents an important alternative for pathogen control and environmental sustainability [10].

Researchers and breeders have long been working to increase plant resistance by focusing their efforts on resistance genes (R-genes), which mediate the recognition of race- or isolate-specific effector proteins of the pathogen, with the subsequent activation of plant defense responses (effector-triggered immunity). However, introgression of resistance genes in elite genotypes is time-consuming and often short-lived, because the widespread deployment of R-genes selects for pathogen strains capable of overcoming plant resistance [7,11]; thus, continuing the discovery and introgression of new R-genes is of prime importance [12,13]. 

To establish a compatible interaction, pathogens use host factors encoded by plant susceptibility genes (S-genes) [14,15,16]. Evidence shows that disabling plant S-genes provides a broad-spectrum and durable type of resistance [17]. For example, elite barley lines carrying introgressed homozygous mutated alleles of an S-gene (*Mildew Locus O, mlo*) have been successfully used in European agriculture for approximately three decades because of the exceptional efficacy and longevity of powdery mildew resistance [18,19]. The *mlo*-based resistance has also been described in several other monocotyledonous and dicotyledonous plant species [20]. Powdery mildew resistant 4 (*PMR4*) has also been identified as an S-gene that no longer supports the normal growth of the powdery mildew pathogen in *Arabidopsis* mutants [21] and has been proved to be a potential candidate gene in disease resistance breeding, because its silencing in tomato and potato did not affect crop growth [12,22]. For obtaining desired mutagenesis, the CRISPR–Cas9 genome editing technology has recently emerged as a revolutionary tool and has been applied in tomatoes since 2014 [23], becoming the primary genome editing tool applied in this species to characterize gene function in precision plant breeding [24]. In particular, the knock-out (KO) of the S-genes in tomato plants led to: (i) *Oidium neolycopersici* susceptibility reduction by *PMR4*-KO [25] and (ii) *Phytophthora infestans* susceptibility reduction by *DMR6*-KO [26]; knock-down (KD) of the *PMR4* S-gene via RNAi silencing led to *Phytophthora infestans* tolerance in potato [22].

In this study, we applied CRISPR/Cas9 editing for disabling the *SlPMR4* gene in two widely cultivated Italian tomato varieties: ‘San Marzano’ (SM), mainly used in the canning industry, and ‘Oxheart’ (OX), highly appreciated for fresh consumption, with the goal to reduce susceptibility to *P. infestans*, the causative agent of LB. The reduction in edited lines’ susceptibility to the pathogen was assessed using a detached leaf assay. Further, we compared the effects of four single guide RNA (sgRNA)-induced mutations and assessed both the overall mutational status and potential unintended off-target effects through Illumina whole-genome sequencing (WGS) of four *SlPMR4* mutants of the SM cultivar.

## 2. Results

### 2.1. CRISPR/Cas9-Based Mutagenesis of SlPMR4

A CRISPR–Cas9 vector containing the *NPTII* resistance gene and four sgRNAs (Figure 1a,b) targeting as many regions of the *SlPMR4* gene was introduced via *A. tumefaciens*-mediated transformation into two commercially available tomato cultivars, SM and OX. Both the cultivars were susceptible to *P. infestans*. A total of 132 SM and 136 OX explants were transformed. Moreover, 87 tomato regenerants (T_0_ generation; 70 in SM and 17 in OX) were obtained. The observed regeneration efficiencies were 65.9% and 12.5% for SM and OX, respectively. From this initial screening, the more robust T_0_ plantlets (26 SM and 9 OX) were recovered from in vitro cultivation, and all of them were positive for *Cas9* PCR amplification (Figure 1c,d). 

Sanger sequencing revealed different editing outcomes for each of the four sgRNAs (Table 1, Appendix A). sgRNA7 was the most efficient, as seven SM (4, 5, 6, 13, 17, 19, and 22) and five OX (2, 3, 4, 9, and 11) genotypes were characterized by more than 99% editing effects, estimated using TIDE. In all cases, the PMR4 gene was fully knocked out due to the introduction of a deletion (−7 bp) in the homozygous state. sgRNA8 also led to the production of 15 SM (4, 6, 7, 8, 9, 12, 13, 14, 17, 18, 19, 22, 24, 25, 26) genotypes with a biallelic mutation (−2 bp, −7 bp) and one plant with a 1 bp insertion present in homozygous status (16). sgRNA8 generated three OX genotypes, with a 2 bp deletion in the homozygous state (2, 4, 11), one with a 1 bp insertion in the homozygous state (3), and one (9) with a biallelic mutation (−2 bp, + 1 bp). The outcome of editing at the sgRNA7 and 8 was the introduction of premature stop codons, which produced shorter truncated proteins. sgRNA1 and 6 showed less efficient results (Appendix A), with the persistence of reference alleles at medium-high frequency and the emergence of few indels in the heterozygous or chimeric state (Appendix A).

Overall, the most frequent indels were small deletions (from 1 to 10 bp, with a predominance of −7 bp and −2 bp) and small insertions (from 1 to 4 bp, with a predominance of +1 bp). For one target (sgRNA7), a 7 bp deletion was observed as the predominant mutation (Appendix A) for both the cultivars. Following Sanger sequencing, we also identified two SM mutants containing large deletions, one of which (SM2) contained a 3200 bp deletion between sgRNA6 and 7 (Appendix A), and the other one (SM5) contained a 146 bp deletion at sgRNA1, although not in the homozygous state.

### 2.2. Reduced Susceptibility to Late Blight (LB)

A DLA was conducted as previously reported [27]. The assay was performed in two independent experiments on 35 edited T_0_ lines (26 SM and 9 OX; Appendix A). In both experiments, the plants with reduced susceptibility showed smaller chlorotic and necrotic foliar lesions than the control plants (Figure 2). In the first experiment, twelve SM and three OX mutants showed reduced disease symptoms (Appendix A), whereas in the second experiment, eight SM and two OX mutants showed a significant reduction in pathogen infection (Appendix A). For further analysis, we selected only mutants that showed significantly reduced symptoms in two independent experiments, including five SM (4, 6, 13, 17, and 19) and one OX (4) mutants (Figure 2, Appendix A). Those mutants showed good TIDE outcomes, which predicted the presence of truncated SlPMR4 proteins in the homozygous state (at the sgRNA7 level), as well as the presence of biallelic deleterious deletions (−2/−7 bp at the sgRNA8 level), generating even shorter truncated proteins. For these reasons, among them, four SM mutants were subjected to WGS for a deeper characterization of the on-/off-target regions. One T_1_ line (SM17-1.2) did not highlight any chimerism (sgRNA7: −7/7; sgRNA1: −2/0; sgRNA8: −7/−2; sgRNA6: +1/0) and similar to the T_0_ plant, showed a reduced susceptibility (Appendix A).

### 2.3. Genomics of Selected Pmr4 Mutants

Four candidate edited lines of the cultivar SM (6, 13, 17, and 19) and one in vitro control plant were subjected to Illumina WGS. Two additional SM WT plants obtained from seedlings germinated in soil were also sequenced. Genome sequencing yielded 1678 billion raw paired-end reads (252 Gb), with an average length of 150 bp (Appendix A). The latter was reduced to 1577 billion (94%) after filtering and trimming high-quality reads. The sequence depth of coverage ranged from 38.1X (SM13) to 52X (SM6), being 46X on average (Appendix A). Sequence data were deposited in the NCBI Short Read Archive with specific submission identifiers (PRJNA846963). 

A de novo genome assembly of each mutant was produced (Appendix A), and blast analysis was used to scan the scaffolds for the presence of T-DNA integration. All four edited plants showed *Cas9* positive scaffolds, and a sequence coverage analysis was used to infer the number of *Cas9* integrations (Appendix A). This analysis highlighted two independent copies in the hemizygous state in each edited plant. The latter was confirmed by qPCR analysis (Appendix A). 

Within these four selected mutants, all candidate gene regions (sgRNA−7, 1, 8, and 6) within SlPMR4 were scanned using CRISPResso2 utilizing WGS data (Appendix A). In all cases, clear evidence of editing was observed (Figure 3, Table 2). Scanning of the *PMR4* in the sgRNA7 region revealed a 100% editing effect and no reference alleles, confirming the TIDE analysis.

In general, the *PMR4* gene was knocked out due to the introduction of a deletion (−7 bp) present in the homozygous state (position 62,314,165–62,314,171 bp in chromosome 7). This mutation can result in a shorter protein lacking 431 amino acids due to the presence of a premature stop codon at position 1337 (instead of 1769 in the WT, Appendix A), thereby affecting the general protein functionality. Scanning of PMR4 in the sgRNA8 region revealed a 100% editing effect with biallelic mutations (a 2 bp deletion at position 62,315,066–62,315,067 in chromosome 7 and a 7 bp deletion at position 62,315,066–62,315,072 in chromosome 7) and no reference alleles, confirming the TIDE analysis. Such mutations can result in shorter proteins lacking 753/751 amino acids, in the presence of premature stop codons at position 1015/1019 (Appendix A), thereby affecting the general protein functionality. When both loci (sgRNA7 and sgRNA8) were affected, the resulting protein originated from a mutation in sgRNA8, which precedes sgRNA7 in the gene (Appendix A). The analysis of *PMR4* at sgRNA1 and sgRNA6 regions showed less efficient editing effects, according to the TIDE analyses (Table 2), with some persistence of the reference alleles and a few indels in the heterozygous or chimeric status (Appendix A). 

### 2.4. Off-Target Events and SNP Analysis

Based on the resequencing data, we evaluated the extent of off-target (OT) mutations caused by CRISPR/Cas9 editing in the four selected pmr4 mutants (SM6, SM13, SM17, and SM19). First, we generated a list of potential OTs (25 loci; Appendix A) for the four sgRNAs that were used to target *SlPMR4*. All 25 candidate OT regions showed sequence similarity with at least 2 bp mismatches with respect to sgRNAs (Table 3), of which seven fell in coding regions and eighteen in non-coding regions (Table 3). For sgRNA6, the number of mismatches increased to five, because no other candidate OTs were observed. The analysis was conducted by mapping the Illumina reads of one control plant, two WT plants, and four mutants to the tomato Heinz 1706 reference genome. All 25 putative OT regions were fully covered by Illumina reads in the control, WT, and *pmr4* mutants (Appendix A), discarding the possibility that large deletions occurred in these plants. A side-by-side comparison of DNA alignments in the control, WTs, and mutants revealed that no SNPs/indels or large deletions were present in candidate OT regions. Indeed, some SNPs/indels were present in the surrounding regions, but they did not indicate any OT effect, being always: (i) conserved nucleotides already in place in SM, but polymorphic with respect to the Heinz 1706 genome; and (ii) outside of the 20 bp window related to the sgRNA-like sequence (putative OT region). In conclusion, we did not find any evidence of mutations in the potential OT regions within the genome of the selected SM *pmr4* mutants. 

Polymorphisms in SM were searched and identified in over seven genotypes, using the Heinz tomato genome as a reference: four selected *PMR4*-edited lines, one control, and two WT materials (Table 4). In total, 595,701 SNPs/indels were observed, with 153,977 cultivars specific for the SM genome with respect to the genome sequence of the Heinz variety. Genotype-specific SNP/indels were identified in all edited lines, as well as in the three unedited plants, to discriminate the emergence of spontaneous mutations (SNP/indels) from mutations induced by in vitro culture or genetic transformation/gene-editing processes. The average SNP number across edited (9.04 SNPs per Mb) and not edited plants (8.95 SNPs per Mb) was comparable, as was the average mutation rate (0.00113% for edited plants, 0.00110% for unedited plants), with no statistically significant differences among them (*t*-test, *p* = 0.78, α = 0.05).

## 3. Discussion

We aimed at investigating whether full KO of an S-gene (*SlPMR4*) through CRISPR/Cas9 editing in two widely cultivated Italian tomato cultivars, SM and OX, may reduce susceptibility to LB, a devastating disease caused by *P. infestans*. We selected four SM *pmr4* mutants and following their whole-genome resequencing, assessed the overall editing efficiency, types of induced mutations, as well as the emergence of any unintended OT effects.

### 3.1. Reduced Susceptibility to LB in Tomato Cultivars Knocked-Out in the SlPMR4 Gene

The S-gene *PMR4* was originally identified because of the powdery mildew resistance phenotype of a *pmr4* mutant in *Arabidopsis* [21]. *PMR4* appears to be the main biosynthetic enzyme that coordinates callose response to biotic, abiotic, and chemical stresses. Even if the callose response is widely recognized as an early response of host plants to microbial attack, callose may protect the fungus during pathogenesis [28,29]. Loss of function of PMR4 results in depletion of callose at fungal penetration sites [21,28] and resistance related to enhanced activation of the salicylic acid signal transduction pathway or constitutive expression of the pathogenesis-related protein 1 (PR-1) [29,30].

In previous studies, *PMR4* mutations have been shown to provide broad-spectrum resistance to powdery mildew [25,28,30]. In this study, we generated loss-of-function mutants in the *SlPMR4* gene in two susceptible tomato cultivars, SM and OX, by applying CRISPR/Cas9 technology. Considering the editing effect on the final phenotype, tolerance to late blight was observed to be dependent on the type of mutation introduced. The three main mutations (sgRNA7-indel−7 bp, sgRNA8-indel−7 bp, and sgRNA8-indel−2 bp) generated trunked copies of the *PMR4* protein, leading to the loss of a large part of the glucane synthase domain (aminoacidic range: 877–1677 (Appendix A) and likely depleting the *PMR4* callose deposition function. The degree of reduced susceptibility to LB was evaluated in the edited mutants using the DLA assay. Compared with the controls, fifteen T_0_ (fourteen SM and one OX) showed reduced susceptibility to *P. infestans* in one experiment, and six of them (SM4, 6, 13, 17, 19, and OX4) showed a reduced susceptibility level in two independent experiments (Appendix A). One edited T_1_ line (SM17-1.2) confirmed a reduction in LB susceptibility analogous to the one detected in T_0_ plants (Appendix A). Our results showed, for the first, time a reduced susceptibility to *Phytophtora infestans* in pmr4 tomato mutants confirming the role of KO *PMR4* in providing broad-spectrum protection against pathogens. In our study, it was also proved that the KO of the *PMR4* gene produced plants with normal growth and with the same habit as that of the WT plants. This was in agreement with the results of previous studies based on observations of *pmr4* mutants [12,21].

### 3.2. Genome Editing Outcomes in Tomato Cultivars

Tomato is a model plant species, and its amenability to transformation methodologies via *A. tumefaciens*, biolistic, or direct protoplast DNA uptake makes it a suitable platform for the application of gene editing technologies [31,32,33]. Tomato is also a food crop, and the improvement of traits achieved by applying gene editing techniques to widely cultivated varieties has the potential for direct use in the field.

In our experiment, we applied CRISPR/Cas9 editing based on the use of 20 nt sgRNAs, which, with respect to longer or shorter sgRNAs, have been reported to be the most efficient in DNA cleavage efficiency. We applied four sgRNAs targeting as many regions of the *SlPMR4* gene, with the goal of increasing the frequency of random insertions or deletions by NHEJ editing [34]. The most frequently induced mutations reported for CRISPR-based experiments are 1 bp and 3 bp deletions, followed by 1 bp insertions [33]. Within the produced mutants, all candidate regions (sgRNA-7, 1, 8, and 6) within *SlPMR4* were scanned using TIDE and CRISPResso2 utilizing WGS data (Appendix A). In all cases, clear evidence of editing was observed (Appendix A, Figure 3, and Table 2), and both TIDE and WGS approaches, showing a high concordance among each other (Table 2), highlighted that small indels at the target sites of sgRNAs were mainly 1 bp insertions (sgRNA6) and 2 bp or 7 bp deletions (sgRNA7, sgRNA1, and sgRNA8; Appendix A). However, as reported in previous studies [25], in some (rare) cases (for example SM2, Appendix A), we also detected the induction of a large deletion (3200 bp). Although it seems difficult to justify the presence of a common repair pattern (e.g., −7 bp), a recent characterization of indel patterns at multiple genomic locations revealed that individual targets show reproducible repair outcomes, with distinct preferences for the class (insertion or deletion) and size of indels [35]. More recently, the role of genetic and epigenetic factors influencing CRISPR−Cas-mediated DNA editing has been clarified by performing large-scale genomic characterization of indel patterns over 1000 sites in the human genome [36]. Therefore, it is now clear that double-strand breaks (DSBs) can be repaired in both predictable and unpredictable manners and that the pattern relies on the target site. Positions −4 and −5 from the PAM seem to roughly predict the likely repair outcome (insertion or deletion). Based on these predictive criteria, we confirmed the tendency of sgRNA6 to introduce an insertion (5 nt preceding PAM=CTGCG-NGG), similar to sgRNA8 (5 nt preceding PAM=ATCAG-NGG) (Appendix A). In contrast, sgRNA7 (5 nt preceding PAM=GGCAA-NGG) and sgRNA1 (5 nt preceding PAM=ACTGG-NGG) showed a tendency to introduce deletions. However, these trends cannot be easily explained if specific patterns of repair in plants are not postulated. Moreover, by analyzing the performance of the sgRNAs and the editing outcome in the two varieties in the study, sgRNA7 resulted in the introduction of a precise deletion and the same type of deletion (−7 bp) in both SM and OX cultivars. These results suggest that by fine-tuning sgRNA at the design stage, it is possible to predict the outcome of gene editing in different plant genotypes. Although the analysis of thousands of repair patterns in plant-specific contexts is needed to provide more details of the CRISPR genome editing outcome and strengthen our prediction power.

Considerable variability in sgRNA efficiency has been detected, which does not appear to change with the expression system or *cas9* delivery method [37,38], and it is often difficult to predict the specificity and stability of sgRNA sequences [39]. In a previous study [25], no mutations were found close to the sgRNA6 target site, suggesting that this sgRNA was not efficient in guiding the *CAS9* protein to induce DSBs. However, in our study, we found that the average efficiency of sgRNA6 was comparable to that of sgRNA1 and lower than those of sgRNA7 and sgRNA8, where no residual reference alleles were highlighted (Figure 3). In this relation, resistance to late blight was dependent on the type/level of editing observed. sgRNA6 was also able to drive the emergence of DSBs in SM2 and generate a large deletion event (Appendix A) of 3200 bp between sgRNA6 and sgRNA7.

### 3.3. Off-Target Absence and Emerging SNPs in Pmr4 Mutants

One of the associated concerns in the application of CRISPR/Cas9 editing is that the endonuclease may act on non-selective and non-specific regions of genomic DNA, commonly known as OT sites [40]. It is believed that the seed regions (8–12 nt most proximal to the PAM) in sgRNA govern the identification of targets, in which a high degree of homology can result in OT binding [41,42,43]. Despite WGS being widely used in plant genomics, studies investigating the occurrence of OTs upon gene editing remain scarce [44,45], and the current studies have primarily highlighted the scarce presence of OT mutations resulting from the *CAS9* activity. In our study, no OT mutations were found (Table 3) in both the coding and non-coding regions in the four SM mutants investigated using Illumina sequencing. Our results confirm that CRISPR/Cas9 can be a highly precise genome editing tool in tomatoes, which is consistent with the results of previous reports [33,46,47,48].

The ratio of spontaneous mutations is highly variable and unique to every organism [49]. Base substitution mutations can often be explained as the result of two main processes: deamination of methylated cytosines and ultraviolet light-induced mutagenesis [50]. Somaclonal variation can further emerge in the conditions of in vitro cultivation, mainly as a consequence of hormone supplementation. In our study, the sequenced mutants (SM6, 13, 17, and 19, Appendix A) were obtained through genetic transformation starting from the WT ‘SM’ (cultivated in vitro). The four mutants showed similar genetic similarity (Table 4), with an average SNP mutation rate of 0.00113%, which was not statistically different from that of the WTs (0.00110%), indicating that the set of SNPs detected in the edited plants did not undergo a statistically significant increase due to editing side effects, as well as somaclonal variation, but arose from spontaneous mutations. Interestingly, the observed spontaneous mutation rate in control tomato (0.00110%) was 5.7-fold higher than that reported in rice [51] and far higher than that reported in *Arabidopsis* [50].

## 4. Materials and Methods

### 4.1. Plant Material and Transformation Vector

Seeds of SM and OX cultivars used in this study were provided by Agrion (www.agrion.it, accessed on 1 October 2022) and maintained in the Germplasm Bank of the Department of Agricultural, Forest and Food Science (University of Torino, Grugliasco, Italy). The vector used for plant transformation was previously reported by Santillán Martínez et al. [25]. It carries the *NptII* resistance gene, *Cas9* gene, and four sgRNA sequences (sgRNA1: GTTAAAGCAGTCCCATACTCG, sgRNA6: GTACTGCCCCACACTCTGCG, sgRNA7: GCCAAGGTTGCCAGTGGCAA, and sgRNA8: GGATATCAGAGAAGGATCAG), designed to target several regions of *SlPMR4* (Solyc07g053980.3.1, ITAG4.1). sgRNA6 targeted the FKS1dom1 domain, while the other three sgRNAs targeted the glucan synthase domain. The four sgRNAs were used to increase the editing efficacy and promote the emergence of deletions between sgRNAs. The transformation vector was cloned into the *Escherichia coli* strain DH5α and then into the *Agrobacterium tumefaciens* strain LBA4404. The transformed *A. tumefaciens* was conserved as a stock at −80 °C and subsequently used for plant transformation, as described below.

### 4.2. Plant Transformation, Regeneration, and Acclimation to Soil

Tomato seeds of both SM and OX cultivars were sterilized with 75% EtOH for 30 s and 1% sodium hypochlorite for 20 min. Sterilized seeds were washed in sterile water for 5 min and then sown on a germination medium (2.2 g/L MS basal salts, 10 g/L sucrose, and 8 g/L Daishin agar; pH 5.8). Seeds were incubated for approximately three days at 4 °C and then transferred at 24 °C for approximately ten days. The expanded cotyledons were used for transformation with *A. tumefaciens*. A single colony of *A. tumefaciens* was streaked in 2 mL Luria–Bertani (LB) medium with antibiotics (50 mg/L rifampicin and 50 mg/L kanamycin) and grown at 28°C for two days, under continuous shaking. The culture was then refreshed in 10 mL LB with the same antibiotics and shaken overnight. The following day, the culture was centrifuged for 15 min at 2000 g, and the pellet was resuspended in liquid induction medium containing acetosyringone (200 mM). The optical density (OD) of the culture was measured at 600 nm (OD600) and adjusted to a final OD of approximately 0.125. After 1 h, the culture was used for transformation. With occasional swirling, the explants were incubated in the culture for 15 min and then incubated at 25 °C in the dark for 48 h. Briefly, cotyledons were cut into four pieces and placed on top of the induction medium (4.3 g/L MS basal salts, 108.73 mg/L vitamins Nitsch, 30 g/L sucrose, 8 g/L micro agar, 1.5 mg/L zeatin riboside, 0.2 mg/L IAA, and 1 mL/L acetosyringone; pH 5.8), with two pieces of filter paper soaked with liquid induction medium (4.3 g/L MS basal salts, 0.4 mg/L thiamine, 100 mg/L Myo-inositol, and 30 g/L sucrose; pH 5.8) supplemented with acetosyringone (200 µM). The explants were then placed in an incubator at 25 °C. Control groups were explants from the wild type (WT) of each variety subjected to the same treatments using a mock solution (liquid induction medium without *A. tumefaciens*). After two days of incubation in the dark, the explants were transferred to a solid induction medium with timentin (300 mg/L) and kanamycin (100 mg/L), and incubated at 25 °C with a 16/8 h light/dark cycle. The medium was renewed every two weeks until shoot proliferation. Two weeks later, the regeneration process was visible, and calli with regenerated shoots were transferred to a shooting medium (3% sucrose, MS + vitamins Nitsch, pH 5.8) supplemented with zeatin riboside (1.5 mg/L), IAA (0.2 mg/L), timentin (300 mg/L), and kanamycin (100 mg/L). Explants from the control group were transferred to the same medium without antibiotics. Once shoots reached 1–2 cm in length, they were transferred to a rooting medium (3% sucrose, MS + vitamins B5, pH 5.8) with or without kanamycin (100 mg/L) and maintained under growth chamber conditions [25,52].

### 4.3. Identification of Transformed Plants and Detection of the SlPMR4 Editing

Genomic DNA was isolated from the leaves of in vitro regenerated plants using the Plant DNA Kit (Omega Bio-Tek, Norcross, GA, USA), analyzed using 0.8% agarose gel electrophoresis, quantified on a Qubit fluorometer (Thermo Fisher, Waltham, MA, USA), and then subjected to PCR screening aimed at amplifying the *Cas9* gene as proof of transgene integration. A positive control (Cas9) was used to amplify a vector containing the *Cas9* gene, whereas a negative control (CTRL) was used on a tomato plant generated in vitro without transformation. PCR was conducted on 5 ng of genomic DNA using the KAPA HiFi HotStart ReadyMix (Kapa Biosystems, Wilmington, MA, USA), according to the manufacturer’s instructions, and Cas9 primers listed in Appendix A. Amplification of the *Cas9* gene was performed using a real-time PCR (RT-qPCR) assay to quantify the number of T-DNA integration events. Reactions were carried out on the same genomic DNA in triplicate, using the Power SYBR^®^ Green Master Mix (Thermo Fisher, Waltham, MA, USA) and the StepOnePlus Real-Time PCR System (Applied Biosystems, Waltham, MA, USA). The following PCR protocol was used: 95 °C for 10 min, followed by 40 cycles of 95 °C for 15 s, and 60 °C for 1 min. Data were quantified using the 2^−ΔΔCt^ method based on the Ct values of *SlActin* (Solyc11g005330.1, ITAG_eugene; primers listed in Appendix A) as the housekeeping gene. The number of *Cas9* integrations was expressed as the relative DNA abundance with respect to single-gene amplification (Appendix A).

To detect the deletions between the sgRNAs, *Cas9*-positive transformants were analyzed by PCR amplification (as reported above) using different primer combinations (Appendix A), flanking the four predicted target gene regions in the *SlPMR4* gene, and compared with control amplicons to detect deletions. Sanger sequencing was performed (BMR Genomics Service, Padova, Italy) on PCR-amplified gene fragments, as described above, using the primers listed in Appendix A. Amplicons of transformants smaller than those detected in control plants, suggesting the occurrence of large deletions between sgRNA targets, were cloned into pGEM^®^-T Easy Vector Systems (Promega, Madison, WI, USA) according to the manufacturer’s instructions, and eight colonies of each transformant were picked and subjected to Sanger sequencing.

Transformants without deletions were PCR amplified using other primers (Appendix A) and then subjected to Sanger sequencing and analyzed using the Tracking of Indels by Decomposition (TIDE, https://tide.nki.nl) assay [53] to quantify the editing efficacy and identify the predominant types of indels (i.e., mutational status). TIDE calculates a goodness-of-fit value (R^2^) as a measure of the reliability of the estimated alleles and the overall efficiency of each TIDE assay as the estimated total fraction of DNA with mutations around the break site.

### 4.4. Detached Leaf Assay with Phytophthora Infestans

*P. infestans* (Westerdijk Fungal Biodiversity Institute strain CBS 120920) was grown on rye sucrose agar medium [54] in the dark at 15 °C. The sporangia were harvested from a 10–14 day old cultured plate by flooding the plate with 5 mL ice-cold water (4 °C) and mixing it properly with a spreader. The plate was maintained at 4 °C for 2–4 h to release the zoospores. Zoospores were harvested by filtering the liquid from each plate through two layers of cheesecloth. Motile zoospores were counted using a hemocytometer under a microscope, and the concentration was adjusted to 2.5 × 10^4^ spores/mL [55].

Six healthy, fully grown leaves from each soil-acclimated line (SM and OX) were used in a detached leaf assay (DLA) according to the procedure described by Foolad [27]. The six leaves of each mutant were randomly placed bottom side up into plastic trays (six replicates) containing water agar (20 g/L). Each plastic tray was divided into eighteen areas, and two groups of mutant leaves and one group of control leaves were randomly placed in each area of the tray. All leaves were infected with the *P. infestans* isolate by dropping a suspension of zoospores at a concentration of 2.5 × 10^4^ spores/mL (12 μL/leaf). After infection, the trays were covered with lids, sealed with Parafilm, and placed in a growth chamber at 20 °C in the dark with a relative humidity of 60%. The trays were examined on a daily basis. Eight days post-inoculation (dpi), images were captured and analyzed using ImageJ software (version 1.52a; LOCI, University of Wisconsin) for the percentage of leaf area damaged (LAD%). A scale of 0–5 was used to score LAD%; a score of 0 indicated the absence of any foliar infection (LAD = 0%), and a score of 5 indicated complete destruction due to LB infection (LAD = 100%; [27,56]. The control leaves were scored and compared to the mutants present in the same tray. The ratio of mutant/control scores was calculated and used to select less-susceptible plants. Statistical differences between mutants/controls were analyzed using a two-tailed *t*-test (*****
*p* < 0.05). Multiple comparisons were performed using a two-tailed Student’s *t*-test with post-hoc Bonferroni correction.

### 4.5. Whole-Genome Sequencing

One microgram of DNA was used to construct short-insert (length 350  bp) genomic libraries (Novogene, Hong Kong), which were sequenced using an Illumina sequencer (Illumina Inc., San Diego, CA, USA) with paired-end chemistry (2 × 150  bp). Raw reads were cleaned with Scythe (v0.991, https://github.com/vsbuffalo/scythe (accessed on 1 October 2022)) to remove contaminant residual adapters and Sickle (v1.33, https://github.com/najoshi/sickle (accessed on 1 October 2022)), which allows the removal of reads with poor quality ends (Q < 30).

A de novo genome assembly was performed using the MegaHit assembler (v1.2.9, https://github.com/voutcn/megahit (accessed on 1 March 2022)), utilizing specific assembly parameters (k-min = 27, k-max = 141, k-step = 10, cleaning-rounds = 1, and disconnect-ratio = 0). Metrics for assessing the quality of a genome assembly (e.g., N50, contig/scaffold number/size/length, and genome length) were obtained using the Perl script Assemblathon_stats.pl (https://github.com/ucdavis-bioinformatics/assemblathon2-analysis (accessed on 1 March 2022)). BLAST analysis was conducted on the assembled genomic sequences (mutants and WTs) to identify any possible insertions using the T-DNA sequence [25] as a query. As preferential choice criteria, the e-value (e-value < 1 × e^−10^), percentage similarity, and query coverage were considered. T-DNA coverage analysis using bedtools (https://bedtools.readthedocs.io (accessed on 1 July 2022)) was conducted to infer the number of *Cas9* integrations within each edited plant by aligning cleaned reads on the T-DNA sequence available as a reference. The *Cas9* copy number was inferred by comparing with the coverage recorded for a single-copy tomato gene (Solyc10g009390.3.1, CYP702). 

### 4.6. On- and Off-Target Analyses and SNP Statistics

In edited plants, the emergence of genomic variants and allele frequencies in the *SlPMR4* locus was highlighted using CRISPResso2 (http://crispresso2.pinellolab.org (accessed on 1 May 2022)) and SNP/indel analysis. Clean reads derived from the edited plants were mapped to the tomato reference genome (SL4.0, https://solgenomics.net (accessed on 1 May 2022)) using the Burrows–Wheeler Aligner (v0.7.17, https://sourceforge.net/projects/bio-bwa/files (accessed on 1 May 2022)) program and ‘mem’ command with the default parameters. BAM files were processed and used for SNP calling using Samtools (v1.9-166-g74718c2) mpileup with default parameters, except for minimum mapping quality (Q  =  20) and filtering out multimapping events (-q  >  1). A variant call format (vcf) file was produced. The vcf file was inspected in the 100 bp window surrounding each sgRNA to highlight SNP/indels through bedtools intersect (https://bedtools.readthedocs.io (accessed on 1 May 2022)).

WT and mutated PMR4 proteins were reconstructed using the ‘getorf’ tool (http://emboss.sourceforge.net (accessed on 1 May 2022)) and proteins were multi-aligned using Clustal Omega (https://www.ebi.ac.uk/Tools/msa/clustalo (accessed on 1 May 2022)). Homology models for the WT and mutated PMR4 proteins were built using the Swiss-Model tool (https://swissmodel.expasy.org (accessed on 1 October 2022)), utilising the AlphaFold-predicted crystal structure of Arabidopsis thaliana PMR4 (PDB ID AF-Q9ZT82-F1), as a template. Pairwise juxtaposition of models was carried out with UCSF Chimera (v1.16, https://cgl.ucsf.edu/chimera (accessed on 1 October 2022)).

For off-target analysis, the CasOT script (https://github.com/audy/mirror-casot.pl (accessed on 1 May 2022)) was used to identify any off-target regions in the tomato genome. All designed sgRNAs were considered as bait in an sgRNA mode, with default PAM type (NGG=A) and specific numbers of permitted mismatches in the seed (2), and non-seed (2) regions allowed. All the candidate off-target genomic region coordinates were intersected with the vcf file through bedtools for editing as well as for the control plants to filter-out monomorphic regions among the latter. The results were analyzed using custom bash scripts.

## 5. Conclusions

Our results showed, for the first time, a reduced susceptibility to *Phytophthora infestans* in *PMR4* tomato mutants confirming the role of *PMR4* knock-out in providing broad-spectrum protection against pathogens. The mutants, subjected to WGS, did not exhibit any evidence of mutations in the candidate off-target regions and the average SNP number was comparable between edited and not edited plants, highlighting a substantial equivalence at the genomic level.

## Figures and Tables

**Figure 1 ijms-23-14542-f001:**
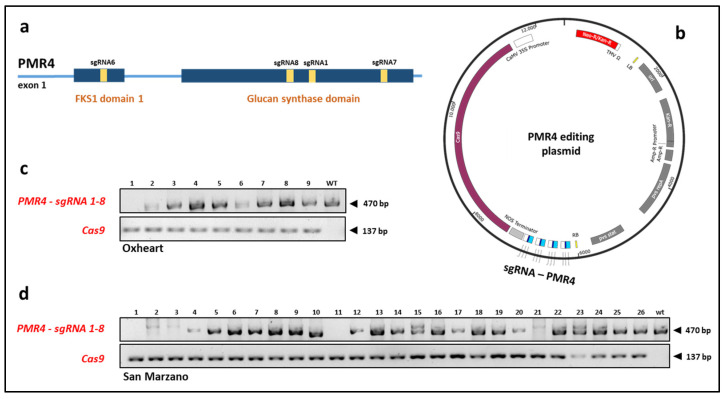
(**a**) Structure of the tomato *PMR4* gene with domains and sgRNAs location (SL4.0-ch07; sgRNA6: 62,317,368–62,317,387; sgRNA8: 62,315,063–62,315,082; sgRNA1: 62,314,925–62,314,944; sgRNA7: 62,315,063–62,315,082). (**b**) Vector containing features for *PMR4* editing through four sgRNAs. Gel electrophoresis of the Cas9 PCR products, as well as the amplicons of the region including sgRNA1 and 8 (PMR4-sgRNA1–8), in (**c**) Oxheart and (**d**) San Marzano tomato mutants.

**Figure 2 ijms-23-14542-f002:**
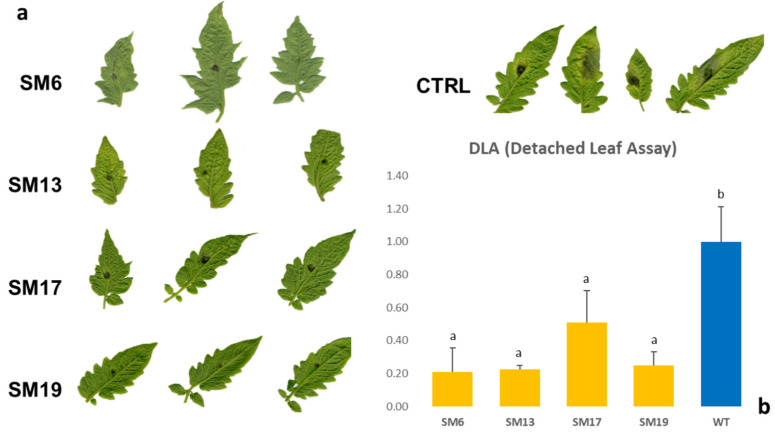
(**a**) Detached leaves assay with *Phytophthora infestans* performed on four *pmr4* San Marzano mutants (SM6, 13, 17, and 19) and a wild type plant as a control group at eight days post-inoculation (dpi). (**b**) In the histogram, normalized LAD% values are reported for each genotype. The y-axis shows the mean ratio of the score of the mutant/control group; bars represent standard deviation (sd). Statistical differences among mutant/control were analyzed with a two-tailed *t*-test (*p* < 0.05). Multiple comparisons were performed using two-tailed Student’s *t*-test with post-hoc Bonferroni’s correction.

**Figure 3 ijms-23-14542-f003:**
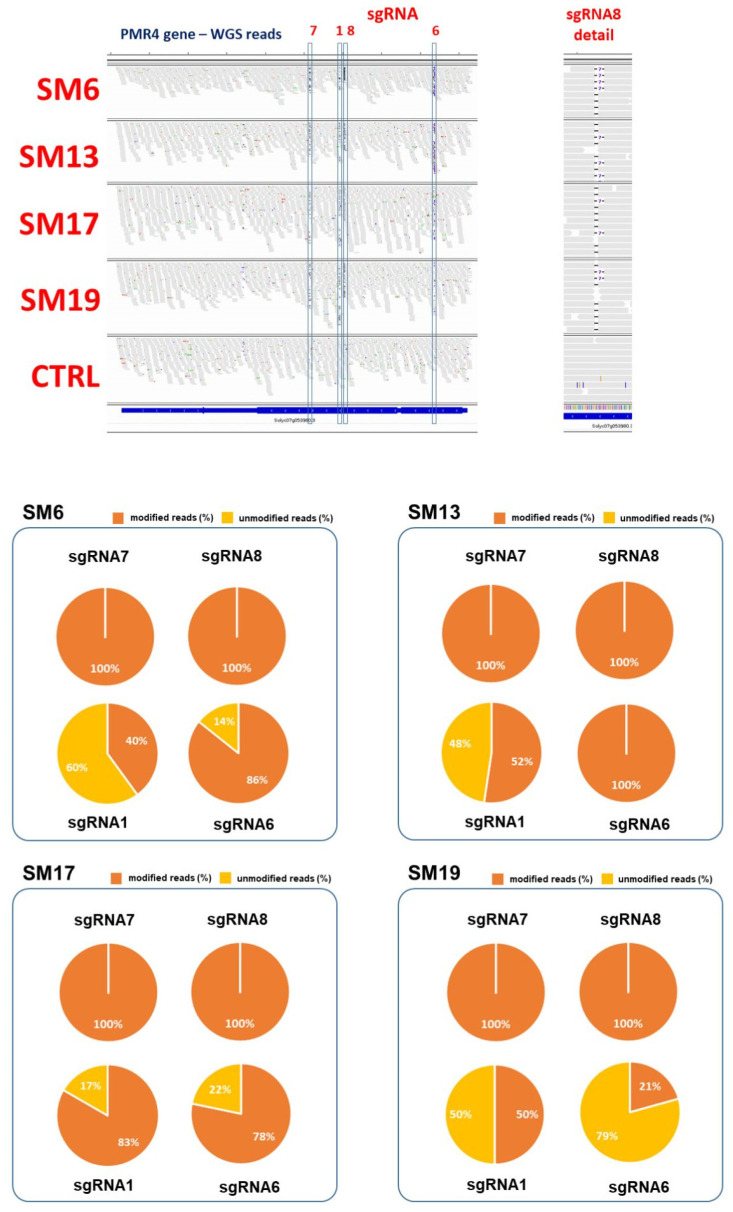
Sequence alignment view of the edited *PMR4* gene at the level of the four sgRNAs in the four mutants and the control plant (top). A focus on sgRNA8 is shown on the right-hand side. Mutational status (%) for each sgRNA region and in all the assayed genotypes as revealed by the CRISPResso2 analysis (bottom).

**Table 1 ijms-23-14542-t001:** Mutational status of the edited genotypes, as revealed by the TIDE analysis of the Sanger sequences. Four regions surrounding the adopted sgRNA were analyzed.

Mutant	Mutational Status (%)-TIDE			
	sgRNA6	sgRNA8	sgRNA1	sgRNA7
**SM1**	1.2	-	-	-
**SM2**	2.5	-	-	-
**SM3**	8.7	-	-	-
**SM4**	38.5	89.4	-	98
**SM5**	24.2	-	-	97.6
**SM6**	40.2	95.1	4.1	98.9
**SM7**	13.1	94.5	58.1	-
**SM8**	34.6	95.5	33.6	-
**SM9**	-	95.4	35.3	-
**SM12**	-	93.3	9.1	-
**SM13**	93.4	90.3	48.1	99
**SM14**	-	92.8	3.4	-
**SM16**	-	97.8	7.6	-
**SM17**	64.6	94.2	76.2	97.9
**SM18**	-	92.9	80.1	-
**SM19**	1.5	93.1	31.3	99.3
**SM20**	1.8	1.1	3.7	2.2
**SM22**	69.3	94.2	9.7	98.3
**SM24**	-	93.6	34.2	-
**SM25**	-	94.2	5.2	-
**SM26**	-	93.9	4.4	-
**OX1**	55.3	-	-	-
**OX2**	17.7	98.6	6.6	99
**OX3**	26.8	98.2	56.7	99.3
**OX4**	14.2	99	9.9	99
**OX9**	13.5	95.9	4.7	99.3
**OX11**	48.6	94.4	32.5	99.3

**Table 2 ijms-23-14542-t002:** Editing effects and observed allelic forms (%) in each of the four sgRNA regions, in four selected genotypes and the San Marzano control. Data were retrieved through: (i) Illumina sequencing analysis, analyzed with CRISPResso2; and (ii) TIDE analysis of Sanger sequences; the overall efficiency of each TIDE analysis is calculated as R^2^.

Mutant			Alleles in sgRNA7 Region (%)											
		R^2^	−18	−10	−8	−7	−6	−5	−3	−2	−1	1	ref	Allelic State
**SM6**	**WGS**	-	-	-	-	100.0	-	-	-	-	-	-	-	**homozygous**
	**Tide**	0.99	-	-	-	98.9	-	-	-	-	-	-	-	
**SM13**	**WGS**	-	-	-	-	100.0	-	-	-	-	-	-	-	**homozygous**
	**Tide**	0.98	-	-	-	98.5	-	-	-	-	-	-	-	
**SM17**	**WGS**	-	-	-	-	100.0	-	-	-	-	-	-	-	**homozygous**
	**Tide**	0.98	-	-	-	97.8	-	-	-	-	-	-	-	
**SM19**	**WGS**	-	-	-	-	100.0	-	-	-	-	-	-	-	**homozygous**
	**Tide**	0.99	-	-	-	97.2	-	-	-	-	-	-	-	
**SM-CTRL**	**WGS**	-	-	-	-	-	-	-	-	-	-	-	**100.0**	**wild type**
**Mutant**			**Alleles in sgRNA1 Region (%)**											
		**R^2^**	**−18**	**−10**	**−8**	**−7**	**−6**	**−5**	**−3**	**−2**	**−1**	**1**	**ref**	**allelic state**
**SM6**	**WGS**	-	-	10.0	-	-	-	-	-	-	23.3		69.0	**chimeric**
	**Tide**	0.96	-	-	-	-	-	-	-	-	-	-	92.2	
**SM13**	**WGS**	-	-	-	-	-	52.4	-	-	-	-	-	47.6	**heterozygous**
	**Tide**	0.92	-	-	-	-	45.7	-	-	-	-	-	43.9	
**SM17**	**WGS**	-	-	-	-	-	-	-	-	62.5	8.3	8.3	20.8	**chimeric**
	**Tide**	0.94	-	-	-	-	-	-	-	54.7	15.0	3.2	18.3	
**SM19**	**WGS**	-	-	-	43.7	-	-	-	-	-	-	-	56.3	**heterozygous**
	**Tide**	0.91	-	-	27.6	-	-	-	-	-	-	-	60.0	
**SM-CTRL**	**WGS**	-	-	-	-	-	-	-	-	-	-	-	**100.0**	**wild type**
**Mutant**			**Alleles in sgRNA8 Region (%)**											
		**R^2^**	**−18**	**−10**	**−8**	**−7**	**−6**	**−5**	**−3**	**−2**	**−1**	**1**	**ref**	**allelic state**
**SM6**	**WGS**	-	-	-	-	33.3	-	-	-	66.7	-	-	-	**biallelic**
	**Tide**	0.95	-	-	-	46.4	-	-	-	48.3	-	-	-	
**SM13**	**WGS**	-	-	-	-	66.7	-	-	-	33.3	-	-	-	**biallelic**
	**Tide**	0.9	-	-	-	43.4	-	-	-	45.8	-	-	-	
**SM17**	**WGS**	-	-	-	-	20.0	-	-	-	80.0	-	-	-	**biallelic**
	**Tide**	0.94	-	-	-	43.2	-	-	-	50.1	-	-	-	
**SM19**	**WGS**	-	-	-	-	50.0	-	-	-	50.0	-	-	-	**biallelic**
	**Tide**	0.93	-	-	-	45.1	-	-	-	47.7	-	-	-	
**SM-CTRL**	**WGS**	-	-	-	-	-	-	-	-	-	-	-	**100.0**	**wild type**
**Mutant**			**Alleles in sgRNA6 Region (%)**											
		**R^2^**	**−18**	**−10**	**−8**	**−7**	**−6**	**−5**	**−3**	**−2**	**−1**	**1**	**ref**	**allelic state**
**SM6**	**WGS**	-	-	-	-	5.7	-	14.3	-	17.1	-	45.7	17.1	**chimeric**
	**Tide**	0.95	-	-	-	11.4	-	12.0	-	16.8	-	-	54.8	
**SM13**	**WGS**	-	25.0	-	-	-	-	3.1	21.9	-	3.1	47	-	**chimeric**
	**Tide**	0.93	27.5	-	-	-	-	12.0	8.1	-	-	38.8	-	
**SM17**	**WGS**	-	-	-	-	-	56.5	-	-	-	-	21.7	21.7	**chimeric**
	**Tide**	0.93	-	-	-	-	31.7	-	-	-	-	31.7	28.2	
**SM19**	**WGS**	-	-	-	-	-	-	-	-	-	-	17.2	82.8	**heterozygous**
	**Tide**	0.98	-	-	-	-	-	-	-	-	-	-	97.0	
**SM-CTRL**	**WGS**	-	-	-	-	-	-	-	-	-	-	-	**100.0**	**wild type**

**Table 3 ijms-23-14542-t003:** Statistics on *PMR4* off-target regions analyzed in all the assayed genotypes and in the San Marzano unedited plants.

sgRNAs	Off-Target All	Coding	Non-Coding	Obs. SNP/Indels
7	4	1	3	0
1	5	2	3	0
8	6	0	6	0
6	10	4	6	0
Total	25	7	18	0

**Table 4 ijms-23-14542-t004:** SNP statistics for each Illumina-sequenced genotype.

Genotype	Plant Type				SNP			
		Genot. Specific	Homoz.	Heteroz.	In Exons	(%)	Per Mbp	Per Mbp (avg)
**SM6**	edited	8141	3318	4823	240	0.0010%	9.6	8.95
**SM13**	edited	9841	3554	6287	348	0.0013%	8.0	-
**SM17**	edited	8589	3491	5098	286	0.0011%	9.1	-
**SM19**	edited	8587	2945	5642	249	0.0011%	9.1	-
**CTRL**	in vitro	7960	2969	4991	236	0.0010%	9.8	9.04
**WT-1**	from seed	8784	2925	5859	269	0.0011%	8.9	-
**WT-2**	from seed	9322	3236	6086	279	0.0012%	8.4	-

## Data Availability

Sequencing data used in this study are openly available in the NCBI database (PRJNA846963).

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
