# Peer review of "CRISPR/Cas9-Based Knock-Out of the PMR4 Gene Reduces Susceptibility to Late Blight in Two Tomato Cultivars"

_ijms, 2022, doi:10.3390/ijms232314542_

Round 1
Reviewer 1 Report
Li R. et al. CRISPR/Cas9-based knock-out of the PMR4 gene reduces susceptibility to late blight in two tomato cultivars
The inciting prospects of breeding pathogen-resistant crops by knocking out their susceptibility genes have already produced several promising reports from the laboratories; however, their agricultural outcome is not yet as evident.
The MS under review describes such knocking out of tomato susceptibility gene SlPMR4 using the CRISPR-Cas9 vector containing four 19 single guide RNAs (sgRNAs; sgRNA1, sgRNA6, sgRNA7, and sgRNA8). These sgRNAs caused mutations with frequencies of 22 to 100%. The researchers demonstrated that these mutations resulted from precise insertions or deletions and mapped them to specific domains of tomato genome.
CRISPR/Cas9 editing may affect non-specifically other domains of genomic DNA. The authors demonstrated the absence of any off-target changes in tomato genome.
Transformed plants were assessed for their resistance to oomycete Phytophthora infestans, the infection agent of late blight disease, in the laboratory test with detached leaves using a specially prepared suspension of oomycete zoospores. In this test, knocking out the SlPMR4 gene reduced plant susceptibility by 55 to 80% depending on particular mutations. These data prove the prospects of CRISPR/Cas9 editing and also presume the possibility of fine-tuning of this technology.
I did not find any evident faults in this MS and readily recommend publishing it ASAP.
Emil Khavkin

Author Response
We thank the reviewer for the positive evaluation.
Reviewer 2 Report
The manuscript investigated the susceptibility of PMR4 knockout mutants of two tomato varieties to late blight using CRISPR/Cas9 system and the four generated lines were sequenced using Illumina whole-genome sequencing for deeper characterization of on-/off-target effects. The manuscript may be interesting, but I found it is quite difficult to follow. For example: 1) What is the meaning of this study? As described in the text, PMR4 knockout mutants have been reported tolerant to powdery mildew in Arabidopsis and tomato; and PMR4 knock-down in potato has been shown to confer tolerance to late blight. The PMR4 knockout mutant even reported resistance against many pathogens such as G. cichoracearum, G. orontii and Hyaloperonospora arabidopsidis. So, the work in this study may be interesting in practice, but it is difficult to see novelty fitting this journal. The vector used to transform is even from the previous study. 2) the author claimed that PMR4 knock-out mutants reduces susceptibility to late blight in the title, but actually, there are many lines are not (Suppl. Table 4), the percentage of non-reduced lines even more than the reduced lines. So how can you get the conclusion that PMR4 knockout mutant reduces susceptibility to late blight? 3) your purpose is to investigate the role of PMR4 against P. infestan, why don’t you just know-out the whole PMR4 gene? The authors took tedious description for various mutants, but none of them is PMR4-knockout fully.
The more specific comments:
Citations: the format of citations in the text are not consistent, mixed by number and author-date
Figure 1: the legend is not clear. a) should be the location of sgRNAs but not the sgRNAs in the PMR4 gene; in addition, why are the sgRNAs not named orderly like 1 to 4? c) need to note what PMR4-sgRNA1-8 is? why are there not negative control in c and d?
Table 1 What is the mutational status? How to get the values?
The susceptibility of mutants to late blight should be confirmed in plants at least in greenhouse.
Author Response
The manuscript investigated the susceptibility of PMR4 knockout mutants of two tomato varieties to late blight using CRISPR/Cas9 system and the four generated lines were sequenced using Illumina whole-genome sequencing for deeper characterization of on-/off-target effects. The manuscript may be interesting, but I found it is quite difficult to follow. For example:
1) What is the meaning of this study? As described in the text, PMR4 knockout mutants have been reported tolerant to powdery mildew in Arabidopsis and tomato; and PMR4 knock-down in potato has been shown to confer tolerance to late blight. The PMR4 knockout mutant even reported resistance against many pathogens such as G. cichoracearum, G. orontii and Hyaloperonospora arabidopsidis. So, the work in this study may be interesting in practice, but it is difficult to see novelty fitting this journal. The vector used to transform is even from the previous study.
AUTHOR’s reply: Thank you for your comment. Indeed, as reported in both introduction and discussion of our manuscript, literature reports information on PMR4 gene. The examples you cited (e.g.: resistance against the pathogens such as G. cichoracearum, G. orontii and Hyaloperonospora arabidopsidis) refers to Arabidopsis PMR4 mutants or potato PMR4 disabling based on RNAi. So far, no data on reduced susceptibility to late blight in tomato following editing of PMR4 gene has been reported in literature, thus our results represent a novelty; this is now better addressed in the manuscript (see lines 31, 288, 538-544). We believe that a further novelty is the development of late blight tolerant genotypes in widely cultivated tomato varieties, confirming the possibility of a direct application of the results we have achieved. Moreover, the previous PMR4 CRISPR mutants were not analysed for on/off targets, and this represents another novelty feature in the paper.
2) the author claimed that PMR4 knock-out mutants reduces susceptibility to late blight in the title, but actually, there are many lines are not (Suppl. Table 4), the percentage of non-reduced lines even more than the reduced lines. So how can you get the conclusion that PMR4 knockout mutant reduces susceptibility to late blight?
AUTHOR’s reply: Thank you for your comment. As expected, we found that the tolerance to late blight was related to the type/level of editing detected. Plants fully mutated, with indels introduced by both sgRNA8 and sgRNA7 (100% editing in those regions, Figure 3b), showed a marked reduction in susceptibility to late blight. These mutations generated trunked copies of the PMR4 proteins (see suppl. Figure 3), leading to the loss of a large part of the glucan synthase domain and likely depleting the PMR4 callose deposition function. Such edited lines showed reduced susceptibility in repeated independent experiments (Table S4) and thus were selected (we apologise, as p-values of Ox plants - 1° and 2° exps - were erroneously reported in supplementary table 4 and now have been reported in the table and manuscript). For this reason, we have discarded the other lines showing a partial editing outcome, with likely incomplete/partial effect on late blight tolerance. In a few cases, the editing outcomes generated a promising reduction of susceptibility, even if not statistically significant (p-value > 0,05). We have modified the manuscript stressing the concept that the tolerance to late blight was dependent on the type/level of editing observed (line 277).
3) your purpose is to investigate the role of PMR4 against P. infestans, why don’t you just know-out the whole PMR4 gene? The authors took tedious description for various mutants, but none of them is PMR4-knockout fully.
AUTHOR’s reply: Thank you for your comments, which have allowed us to improve the clarity of the results reported in our manuscript. We used 4 sgRNAs to maximize the editing efficiency and targeting separately two distinct domains: (i) FKS1dom1 and (ii) Glucan-synthase domain. In our four selected mutants, we obtained a full PMR4-knockout (Table 2, Figure 3) at both sgRNA7 and sgRNA8 level. The induced mutations were homozygous and bi-allelic (following resequencing no traces of reference alleles were detected), and we believe had a great impact on protein functionality being the Glucan-synthase domain depleted due to the presence of a premature stop codon (shorter proteins were present as highlighted in Suppl. Figure 3).
We apologize for the formatting of Table 2, as the last column was only partially readable, making the reported data hard to understand. The table 2 has been amended.
The more specific comments:
REVIEWER 2
Citations: the format of citations in the text are not consistent, mixed by number and author-date
AUTHOR’s reply: According to the referee’s comment, we have modified them in the text.
REVIEWER 2
Figure 1: the legend is not clear. a) should be the location of sgRNAs but not the sgRNAs in the PMR4 gene; in addition, why are the sgRNAs not named orderly like 1 to 4? c) need to note what PMR4-sgRNA1-8 is? Why are there not negative control in c and d?
AUTHOR’s reply: Thanks to the referee for the comment. a) We have now modified the legend, accordingly. In the manuscript we named the sgRNA in the same way they have been indicated in an our previous study (Santillán Martínez, 2020), in order to avoid misunderstandings. The sequence of sgRNAs can be found in the Materials and Methods. We have now explained in the legend the meaning of “PMR4-sgRNA1-8”, the amplification of the region including sgRNA1 and 8. In this experiment, we considered, as negative control, the wild type plants since they do not contain any CAS gene. We routinely perform negative PCR control (water), although not reported in the Figure 1c,d.
REVIEWER 2:
Table 1 What is the mutational status? How to get the values?
AUTHOR’s reply: We thank the referee for addressing this issue. The TIDE software quantifies the editing efficiency and identifies the predominant types of insertions and deletions (indels) in the DNA of a targeted cell pool, reported as percentage of genomic region edited in each genotype. It is defined as “mutational status” and it is linked to editing at sgRNA regions). We have now better specified this in the materials and methods (line 454).
REVIEWER 2:
The susceptibility of mutants to late blight should be confirmed in plants at least in greenhouse.
AUTHOR’s reply: We did perform the cultivation of edited and wt plants in greenhouse, whose leaves were then analyzed with DLA. We applied the detached leaf assay (DLA), since it is a nondestructive method for evaluating interactions between plants and disease-causing agents and it allows a quick assessment of the potential pathogens' infectivity.
Reviewer 3 Report
The Ms ‘CRISPR/Cas9-based knock-out of the PMR4 gene reduces sus- 2 ceptibility to late blight in two tomato cultivars’ is very nicely designed, executed and written. Authors have covered almost all the aspects of the study.
It can be accepted for publications. I have only a few queries-
1. Did authors analysed the trans gene free edited lines? It will further increase the value of Ms.
2. What is the reasons behinds SNPs in different lines, if those regions are not the offtarget?
Author Response
Did authors analysed the transgene free edited lines? It will further increase the value of Ms.
AUTHOR’s reply: Thank to referee for the comment. Unfortunately, we have not analyzed transgene-free lines as they were not available.
What is the reasons behinds SNPs in different lines, if those regions are not the offtarget?
AUTHOR’s reply: Thank to the referee for the comment. We evaluated genotype-specific SNP/indels in all edited lines, as well as in three not edited plants, with the aim to compare the frequency of spontaneous mutations (SNP/indels) and the frequency of mutations potentially induced by in vitro culture or genetic transformation/gene-editing processes. The average SNP number between edited and not edited plants was comparable, and thus it was considered to represent the ‘average mutation rate’. Our results highlight the substantial equivalence in the frequency of genome mutations between edited and not edited lines.
Reviewer 4 Report
Authors submitted manuscript entitled “CRISPR/Cas9-based knock-out of the PMR4 gene reduces susceptibility to late blight in two tomato cultivars” to IJMS. In this study, authors knocked out PMR4 gene in two tomato cultivars. They found that mutants showed resistance to Phytophthora infestans. Moreover, they perform genotyping of mutants and fund several homozygous, bi-allelic and mono-allelic mutants. In addition, they also performed whole genome sequencing to screen off-target effects. This study is very interesting, and results are novel and valuable. I’ve read this manuscript carefully and found that it is very well written and falls to aims and scopes of the journal.
I suggest authors to add few concluding lines in the end of discussion and try to improve this section.
It is better if authors can insert of pictures of whole plants (WT and mutants).
This manuscript is very well designed and acceptable for publication.
Author Response
Thank to the referee for the comments. Accordingly, we have modified the manuscript, by adding a conclusion paragraph. We have also inserted a picture of the mutants and wt plants as supplementary figure 5.
Round 2
Reviewer 2 Report
accepted
Reviewer 3 Report
Authors have addressed all thw queries.